# Scalable Screening and Treatment Response Monitoring for Perinatal Depression in Low- and Middle-Income Countries

**DOI:** 10.3390/ijerph18136693

**Published:** 2021-06-22

**Authors:** Ahmed Waqas, Abid Malik, Najia Atif, Anum Nisar, Huma Nazir, Siham Sikander, Atif Rahman

**Affiliations:** 1Institute of Population Health, University of Liverpool, Liverpool L69 3BX, UK; sihamsikander@gmail.com (S.S.); Atif.Rahman@liverpool.ac.uk (A.R.); 2Human Development Research Foundation, Islamabad 44210, Pakistan; abid.malik@hdrfoundation.org (A.M.); najia.atif@hdrfoundation.org (N.A.); huma.nazir@hdrfoundation.org (H.N.); 3School of Nursing, Xi’an Jiaotong University, Xi’an 710061, China; Anum.nisar@hdrfoundation.org; 4Health Services Academy, Islamabad 44000, Pakistan

**Keywords:** Patient Health Questionnaire-9 (PHQ-9), Patient Health Questionnaire-4 (PHQ-4), Hamilton Depression Rating Scale, assessment, diagnosis, perinatal depression, maternal depression, screening, treatment response

## Abstract

Common perinatal mental disorders such as anxiety and depression are a public health concern in low- and middle-income countries. Several tools exist for screening and monitoring treatment responses, which have frequently been tested globally in clinical and research settings. However, these tools are relatively long and not practical for integration into routine data systems in most settings. This study aims to address this gap by considering three short tools: The Community Informant Detection Tool (CIDT) for the identification of women at risk, the 4-item Patient Health Questionnaire (PHQ-4) for screening women at high-risk, and the 4-item Hamilton Depression Rating Scale (HAMD-4) for measuring treatment responses. Studies in rural Pakistan showed that the CIDT offered a valid and reliable key-informant approach for the detection of perinatal depression by utilizing a network of peers and local health workers, yielding a sensitivity of 97.5% and specificity of 82.4%. The PHQ-4 had excellent psychometric properties to screen women with perinatal depression through trained community health workers, with a sensitivity of 93.4% and specificity of 91.70%. The HAMD-4 provided a good model fit and unidimensional construct for assessing intervention responses. These short, reliable, and valid tools are scalable and expected to reduce training, administrative and human resource costs to health systems.

## 1. Background

Perinatal depression (PND) is a public health concern in low and middle income countries (LMIC), especially Pakistan, where its prevalence is estimated at 32.4% (95% CI: 25.4–40.3%) [1]. It is associated with both poor maternal physical and mental health outcomes and poor child health [2]. Perinatal depression is associated with poor maternal quality of life, social and marital relationships, and also hampers mother-child relationships [2]. Among their newborns, mothers with PND report poor anthropometric measures, physical health, poor sleep, motor, cognitive, language and socioemotional development [2,3,4]. Thus, children born to mothers with PND suffer from inequities beginning at the very start of their lives, contributing to a vicious intergenerational cycle of poverty and socioeconomic inequity [5]. The long-term economic costs of common perinatal mental health problems amount to about GPB 8.1 billion for each one-year cohort of births in the UK [6], and is likely to be higher in countries like Pakistan, given the associations with poor child growth [7,8].

Recognizing the public health consequences of poor maternal and adolescent mental health, mental health care was established as a priority area by the President of Pakistan, in 2019 [9]. This Mental Health Programme aims to leverage information technology for nationwide implementation of psychosocial interventions to promote mental health and prevent mental illnesses among mothers and children. For perinatal mental health, the WHO Thinking Healthy Programme (THP) for perinatal depression is being implemented at the primary care level, which is a low-intensity multicomponent cognitive behavioral intervention that can be delivered by community health workers [10]. However, to ensure the success of such interventions, an appropriate mechanism for the detection and screening of PND in community settings is urgently required for implementation purposes.

Several tools exist for this purpose, which have frequently been tested in clinical and research settings globally [11,12,13,14,15]. However, these tools require patients to respond to multiple items on a questionnaire and are often designed with a literate population in mind [16]. Although these tools are implementable in a limited clinical or research setting, they are resource intensive from a population perspective, requiring huge investments in the training of allied healthcare professionals in the correct delivery of screening, as well as time costs. These substantial investments associated with community level screening of perinatal depression [17] are not possible in most LMIC. Brief tools requiring less intensive human resources and training, but demonstrating good psychometric properties are required, especially as the evidence base for effective treatments delivered through primary care health systems grows and scale-up initiatives are implemented [9,18].

Researchers have attempted to create shorter versions of tools for assessing PND. Notable examples of these tools include the 5-item Edinburgh Postnatal Depression Scale (EPDS), which perform comparably well to their full versions [16,19]. Such tools continue to be developed and tested around the world including in Pakistan [15,20]; however, there is a lack of such studies from LMIC.

This study utilizes data curated from three well-designed interventional and epidemiological cohorts to: (a)Propose a two-phased process for perinatal depression identification and screening, integrated into the primary health care system in rural Pakistan.(b)Propose a short tool for measuring the response to treatment for perinatal depression.(c)Report psychometric data on three short tools, including the model fit and accuracy: The Community Informant Detection Tool (CIDT) for identification of high-risk women, the 4-item Patient Health Questionnaire (PHQ-4) for screening for perinatal depression, and the 4-item Hamilton Depression Rating Scale (HAMD-4) for measuring responses to treatment.

## 2. Methods

### 2.1. Approach

We propose a two-phased detection and screening process for perinatal depression in Pakistan (Figure 1). We present evidence for two tools for this purpose: The Community Informant Detection Tool (CIDT) for the identification of high-risk women by community health workers, as well as peers comprising the social network of women, acting as key informants for women being assessed. This would be followed by evaluation with the 4-item Patient Health Questionnaire (PHQ-4) administered by community health workers to all women identified as positive through the CIDT. As a third phase, for measuring treatment responses, we propose a short version of the Hamilton Depression Rating Scale, comprising 4 items (HAMD-4) and administered by community health workers during treatment (Table 1).

### 2.2. Settings and Overall Study Design

For the validation of these psychometric tools, secondary analyses were conducted on three datasets. These datasets were curated from three studies conducted in Kallar Syedan, a rural subdistrict in the Punjab province in Pakistan [10,21,22]. This subdistrict comprises a rural, demographically stable and socioeconomically homogenous area. It comprises 11 administrative units called union councils (UCs), where each UC is served by a primary health care unit (BHU). Besides doctors and medical technicians, each BHU is also served by 122 community health workers called lady health workers (LHWs) who provide basic maternal and child health in their respective areas. This area is also of primary interest because the WHO Mental Health Gap Action Programme (mhGAP) is being rolled out as a pilot program in this area and it is the site of initial implementation of the President’s Programme [9,23]. Details on the studies utilized for this report can be read in detail, in the primary publications [21,24,25]. The three datasets reported in this study provided analyses for each of the tools namely the CIDT [24], PHQ-4 [21] and HAMD-4 [25].

#### 2.2.1. Phase 1: Identification of Women at High Risk of Perinatal Depression

The CIDT approach for identification of probable cases of PND utilizes networks of key informants in communities [25]. The development, testing and validation of our tool, based on this approach and adapted specifically for use by community health workers (CHWs) and peers in primary care settings in Pakistan, has been explored in greater detail elsewhere [24]. This approach was envisaged as a proactive, community driven case identification program in a task shifting framework, for detection of women suitable for low- intensity mental health interventions. It has been shown to be effective in supporting the constrained primary health care system in resource poor areas in Pakistan [24], and found suitable for use by lay community informants.

The CIDT tool comprises three major sections: (a) status of pregnancy and experience of social stressors, (b) illustrations for PND symptoms to aid in visual recognition by the informants (rated as yes/no), and (c) visual analog for the likelihood of a match. The CIDT tool presents illustrations of nine PND symptoms: (i) difficulty in sleeping, (ii) loss of appetite, (iii) agitation, (iv) lack of concentration, (v) helplessness, (vi) fatigue, (vii) loss of interest, (viii) low mood, and (ix) suicidal ideation. The last section comprises a point visual analog ranging from (i) no match, (ii) some match, (iii) most match, to (iv) complete matching. These were developed as illustrative vignettes to facilitate prompt recognition of PND cases; after thorough formative research to ensure the acceptability of the language used and adequate sensitivity and specificity levels. The tool provided a simple one-page illustrated guide for LHWs to identify pregnant women with such features from their own knowledge or to ask another key informant known to the woman. We found the CIDT was preferred over traditional instruments such as the PHQ-9 and EPDS in rural settings due to barriers pertaining to poor literacy and administrative time constraints.

For the CIDT tool, a cross-sectional study was conducted in the Kallar Syedan region, where 44 community health workers and 18 local women in three Primary Health Care Center (PHC) catchment areas were approached to participate in the study as community informants. Those consenting to participate in the study were given a two-hour psychoeducational session on maternal depression. The participants were trained in four important steps of detection of PND using the CIDT tool: identification, matching to illustrations, assessment of day-to-day functioning, and help-seeking behaviors. Thereafter, data collection was started in the community over four months (June to September in 2019), where the informants carried out active case detection of probable positive and negative cases for PND. All pregnant women or mothers of children up to three years old in the area, identified from PHC records, were potential targets for detection of PND.

#### 2.2.2. Phase 2: Screening of High-Risk Women for Perinatal Depression

For second-phase screening (or confirmation) of PND, we examined the 4-item Patient Health Questionnaire. This questionnaire comprises two items each for depressive symptoms and anxiety symptoms. The depressive symptoms correspond to a sad mood and anhedonia while the anxiety symptoms assessed were nervousness and a lack of control on worrying. This scale was chosen because it is brief, assesses for depression and comorbid anxiety symptoms, and has shown sensitivity and specificity comparable to the original PHQ-9 for the screening of PND [12,26,27]. The assessment of comorbid symptoms of anxiety was of particular interest because of the transdiagnostic nature of the community-based perinatal depression interventions, such as the WHO Thinking Healthy Programme, as well as the high prevalence of comorbid anxiety symptoms among women with PND [14,15,25].

The data relevant to the PHQ-4 were curated from a larger prospective birth cohort—the Bachpan cohort—set up in Kallar Syedan [21]. This birth cohort was conducted between October 2014 and February 2016, recruiting a total of 1154 pregnant women from Kallar Syedan. A team of trained research assistants performed a battery of psychometric tests pertaining to maternal psychosocial measures and child growth, cognitive, and socioemotional measures [21]. Follow-up assessments were performed at the child’s 3rd, 6th, 12th, 24th and 36th month of age. The instruments of interest in this dataset were the Patient Health Questionnaire (9-item) and the Generalized Anxiety Disorder-7 scale. These two instruments are longer versions of the PHQ-2 and the GAD-2, which together form the PHQ-4. Further details on the cohort can be found elsewhere [21].

#### 2.2.3. Phase 3: Measuring Treatment-Response

For the measurement of treatment-responses, a brief version of the Hamilton Depression Rating Scale (HAM-D) was considered [28]. HAM-D is a popular and validated tool that has been employed worldwide in psychotherapeutic research for measuring responses to interventions and found to be a valid and reliable instrument in previous research [10,28,29,30]. However, recent studies have indicated that the original version of the HAM-D may be less sensitive to change from varied psychotherapeutic and psychopharmacological interventions [26]. Therefore, an adapted and shorter version of the HAM-D, comprising six items pertaining to six core depressive symptoms has been proposed as having a better sensitivity to change [26]. These core symptoms together form the HAMD-6 or the melancholia subscale of HAMD-17. These core symptoms exhibit a unidimensional construct and pertain to depressed mood, guilt, work and activities, psychomotor retardation, psychic anxiety, and general somatic symptoms (energy and physical pain) [26]. In addition, another short version comprising four items of HAM-D corresponding with the PHQ-4 has been considered [26]. These four items correspond to four depressive and anxiety symptoms on the HAM-D: sad mood, anhedonia (lack of interest and enjoyment), psychic anxiety and somatic anxiety.

For our analyses, we curated the data pertaining to the HAM-D from a cluster randomized controlled trial (cRCT) conducted in Kallar Syedan and Gujar Khan. This cRCT was designed to test the effectiveness of the WHO Thinking Healthy Programme compared to treatment as usual. Recruitment to this trial was done from April 2005 to March 2006. All those consenting women were recruited who were in their third trimester of pregnancy, aged 16–45 years, married, and resident of the respective UCs. More details on this study design can be read elsewhere [25].

## 3. Statistical Analyses

All analyses were conducted in SPSS (v. 25) and AMOS (v. 25). Quantitative variables were presented as the mean (SD) and categorical frequency (%). For the CIDT, validation results were described from the original study published elsewhere [24]. For PHQ-4 and HAMD-6, confirmatory factor analyses in a structural equation modelling framework were conducted to assess the model fit of the PHQ-4 and HAMD-6. Several indices for goodness of fit were used: *χ*^2^*/df,* root mean square error of approximation *(RMSEA), standard root mean squared residual (SRMR), comparative fit index (CFI), normed fit index (NFI), Tucker-Lewis index (TLI), incremental fit index (IFI), goodness-of-fit index (GFI), and adjusted goodness-of-fit index (AGFI)* [31]. Cut-off values for goodness of fit indices were ≥0.90 for CFI and TLI, 0.05 to 0.10 for RMSEA, and >0.80 for AGFI [31]. Criterion validity of the PHQ-4 and the HAMD-6 were conducted using the Area under the curve analyses. The comparator gold standard used in these analyses was the clinical diagnosis obtained using The Structured Clinical Interview for DSM-5 (SCID module) by trained psychiatrists and psychologists [32].

## 4. Results

### 4.1. Phase 1: Community Assessment with the Community Informant Detection Tool

Detailed analyses for the CIDT tool have been reported in a previous publication [24]. Briefly, the CIDT tool was tested in a cross-sectional sample of 435 women who underwent screening by community health workers and peers, with a mean age of 28 (4.7) years. Over 90% of the women had education levels between the eighth and tenth grade. A majority (73.2%) had one to three children and belonged to a low or lower-middle socioeconomic class, with salary levels below PKR 30,000 per month (*n* = 287, 65.98%).

Using the CIDT tool, there were 150 probable positive cases and 275 probable negative cases as identified by the presence of at least one symptom of PND, by the informants. The CIDT tool presented an excellent internal consistency with Cronbach’s alpha values for illustrations of the nine symptoms of depression (0.92) and matching to the five pictorial illustrations (0.87). The ICCs for the inter-data collector reliability of CIDT-MD ranged from 0.87 to 0.89.

Factor analyses revealed moderate to strong communality values (0.32 to 0.71) and factor loadings (0.57 to 0.85). Using the complex scoring method (summing of item scores), a cut off value of 3.5 on CIDT-MD was revealed as having a good sensitivity value of 81% and specificity of 69.4%. Whereas using a complex scoring method, sensitivity of CIDT-MD was revealed to be 97.5% (95% CI: 94.2–99.1) and specificity was 82.4% (95% CI: 77.8–86.4). The tool also distinguished between women with social stressors and their counterparts [24].

### 4.2. Phase 2: Formal Screening with the Patient Health Questionnaire-4 Items

The dataset pertaining to the PHQ-4 items was curated from the *Bachpan* cohort [21]. This cohort comprised 1154 women with a mean age of 26.7 (4.5) years. Mean education levels were reported as 7.7 (4.5) years. Most of the respondents were housewives (93.80%), with an equal class representation from a low socioeconomic class to an upper socioeconomic class. Most of the women lived in joint households (65.80%), with 30.20% reported as having their first pregnancy.

Mean scores on PHQ-4 were 2.41 (3.27), with scores ranging between 0 and 12. For testing the dimensionality of the PHQ-4, no exploratory factor analyses were conducted. Only confirmatory factor analyses with the maximum likelihood method were run to ascertain the model fit for the PHQ-4. A unidimensional model for the PHQ-4 showed an excellent model fit. The CMIN/df was within a permissible range (4.28) while all the absolute fit and relative fit indices were >0.90, indicating an excellent fit: GFI = 99.7, AGFI = 96.6, PGFI = 0.100; NFI = 0.997; RFI = 0.982, TLI = 0.986; CFI = 0.998. RMSEA with a value of 0.07 and PCLOSE of 0.206 (Figure 2).

The PHQ-4 also revealed an excellent criterion validity with an AUC value of 0.967 (Figure 3) at a cut off value of 4.5, yielding a sensitivity of 93.4% and specificity of 91.70%. In addition, PHQ-4 scores also yielded a strong association with total scores on PHQ-9 (r = 0.866, *p* < 0.001) and GAD-7 (r = 0.873, *p* < 0.001).

### 4.3. Phase 3: Measuring Intervention Response with the Hamilton Depression Rating Scale–4 Items

The available data from 903 participants had a higher proportion of women aged between 20 and 29 years (62.46%). Only 3.65% of the study sample was educated above tenth grade while the rest were uneducated (41.42%) or educated till tenth grade. Around 59.80% of the participants lived in joint families, with 21.82% ranked as belonging to a poor socioeconomic class.

Mean scores on the HAMD-4 were 7.35 (2.30), with scores ranging between two and 13. Firstly, a series of confirmatory factor analyses were run for the HAMD-6 and the HAMD-4. Both the models yielded good model fit, with the HAMD-4 outperforming the HAMD-6 on all the absolute, relative, and comparative indices (Figure 4, Table 2). Therefore, we chose the HAMD-4 for further analyses. The HAMD-4 yielded similar effect sizes postintervention when compared with the HAMD-17. It yielded an effect size (g) of −0.64 (0.07) when compared with the HAMD-17 with −0.63 (0.07). When compared with the SCID postintervention, it revealed an excellent criterion validity. The area under the curve was estimated at 0.948, and at a summed score of 4.5, it yielded a sensitivity of 91.2% and a specificity of 92.5% (Figure 5 and Appendix A). Cronbach’s alpha value for the HAMD-4 was adequate at 0.73.

## 5. Discussion

### 5.1. Summary

The present study presents validity and reliability estimates for three brief scales used for identifying risk, screening, and evaluating the treatment response for perinatal depression. These tools have the potential to be feasibly embedded within the Primary Care system in Pakistan. Moreover, these tools provide excellent comparability and criterion validity when compared with gold standard diagnostic interviews. The use of shorter scales may reduce economic costs to the health system, associated with training of the workforce and administration in clinical and community settings.

### 5.2. The Need for Short Scales for Assessing Perinatal Depression

The recent guidelines on screening for maternal depression by the United States Preventive Services Task Force in the USA and The National Institute for Health and Care Excellence in the UK, recommend use of screening scales PHQ-9 or the EPDS [19,33]. These tools have been thoroughly researched in a variety of settings globally. However, the administration of these scales on a wider scale requires substantial infrastructural and training costs to health systems [17]. There is also resistance from implementers in LMIC to integrate these tools into routine data collection systems due to the relatively large number of items.

### 5.3. Comparisons with Previous Literature

Therefore, this decade has seen a rise in research on short tools for the assessment of mental health conditions. This body of research has shown that the performance of these short tools may be statistically equivalent to their original and longer versions. For instance, Ishihara et al., validated the Patient Health Questionnaire-Depression-4 among 7850 English speaking populations, and found it to be statistically equivalent to its original version (PHQ-9) [11]. It yielded a Cronbach’s alpha of 0.805 and an optimal sensitivity (0.788) and specificity (0.837). Smith et al., have also shown other versions of the PHQ-9 including the PHQ-8 and PHQ-2 to be valid and reliable tools for screening perinatal depression [27]. Smith et al., reported that the PHQ-8 had a sensitivity of 77% and a specificity of 62% while the PHQ-2 had a sensitivity of 62% and a specificity of 79%. However, there have been concerns regarding replication of these findings in different studies, partly due to the heterogeneity in choice of study setting and study population. And meta-analytical research especially by Thombs et al., [16] has shown some evidence for overestimation of perinatal depression when short versions of the EPDS are used.

The present study proposes a two-phase screening process. This may be more feasible in low resourced health systems, especially in close-knit rural areas. CIDT can provide rapid identification of women at risk through key informants, which can be confirmed through the PHQ-4. This approach has been tested and found feasible in some settings [34] exploring a two-step screening program for perinatal depression. Following a two-step approach, Yawn et al. [34] implemented a multistep depression screening program utilizing postal EPDS followed by physician administered PHQ-9, for the assessment of perinatal depression in the US. It found a multi-step approach for screening of perinatal depression among women to be effective in improving rates of diagnosis, treatment seeking behavior and lowering depressive symptom levels. Moreover, these programs required modest resources for implementation because of a funneled approach for screening; using a nontechnical method followed by one requiring a health professional for confirmation of diagnoses.

### 5.4. Limitations of Our Work

Measurement of treatment responses, especially in settings where non-specialists provide care, is important both to monitor individual responses, and to aggregate health system specific data pertaining to program effectiveness. Although shorter versions such as the HAMD-6 has in previous research [26] been shown to be responsive to change, questions have arisen regarding the content validity for ultra-brief scales to measure responses to interventions [35]. This is particularly important because a good scale should address all clinically relevant issues exhibited by the patients. However, ultrashort scales lack in this facet of monitoring responses in detail. For instance, among the participants in this study, symptoms such as somatic problems and insomnia were highly prevalent. These important symptoms are missed while using short scales. Moreover, this consideration is also important because some symptoms may be markers of a complex illness and associated with a different treatment trajectory [36,37,38].

This is also demonstrated in emerging research on mental disorders from a network perspective [38,39,40,41,42]. For instance, Fried et al., has demonstrated conditions such as depression to be very heterogenous and to demonstrate high comorbidity with other mental disorders, especially anxiety [38,39,40,41,42]. This also results in a varied clinical presentation, and impairment in daily activities across individuals [38,39,40,41,42]. Moreover, patients with depression also present with heterogeneous combinations of symptoms that may or may not be covered by the DSM criteria of diagnoses [41]. Despite all of these limitations, clinical research and implementation science has to meet midway, to achieve impact in real world settings.

## 6. Strengths and Implications for Future Research

The present study proposes a three-phased approach for the identification of risk, screening, and a treatment-response system for perinatal depression for large-scale implementation in Pakistan. Three very brief instruments have been successfully integrated into the primary care system, and their validity and reliability indices were demonstrated in relatively large community cohorts with an excellent study design. This approach can be replicated in other low- and middle-income countries.

Despite these strengths, the present research has several limitations which need to be addressed through further research. The present study does not provide data pertaining to cost-effectiveness with the use of different scales. More research needs to be conducted on the economic benefits of using short tools for community level assessments of depression. The present study demonstrates the use of this approach in research settings, reporting the psychometric properties of the short tools. However, the implementation and effectiveness of these tools needs to be explored in real-world settings (including clinical settings). This should ideally be done using implementation research study designs, where these screening tools are evaluated further for implementation outcomes, including penetration and reach in communities and clinical effectiveness for improving depression. The latter point has been demonstrated in the PRIME program in Nepal where use of detection tools led to relatively low detection rates of mental disorders by community health workers [18]. In addition to testing of effectiveness and cost-effectiveness of tools in real work settings, it would also be important to set up supervision and quality assurance procedures for training healthcare workers [18].

A detection and screening program for perinatal depression can only be as effective as the treatment program that is coupled with it. The President’s Programme in Pakistan, if effectively delivered, can improve equity to mental health care, by ensuring mental health treatment for all [9]. To improve equity in mental healthcare, task-shifting is a necessity for low- and middle-income countries. However, more practical steps need to be taken in designing feasible tools for mental health by non-specialist healthcare personnel. One of the tools suggested in this study, the HAMD-4, although comprising of only four items, still requires ample training in its administration due to the inherent objective assessment of symptoms. Besides using easy to understand tools, it is necessary to develop easy to administer response systems as well. We recommend that future studies should test the feasibility of visual analogue responses, illustrative vignettes, and easy to administer Likert scale responses, in testing sensitivity to change after delivery of an intervention. Another important step for future research is to test these tools in other provincial in languages other than Urdu, Potohari and Punjabi, to ascertain their measurement invariance across different provinces and ethnicities in Pakistan.

## Figures and Tables

**Figure 1 ijerph-18-06693-f001:**
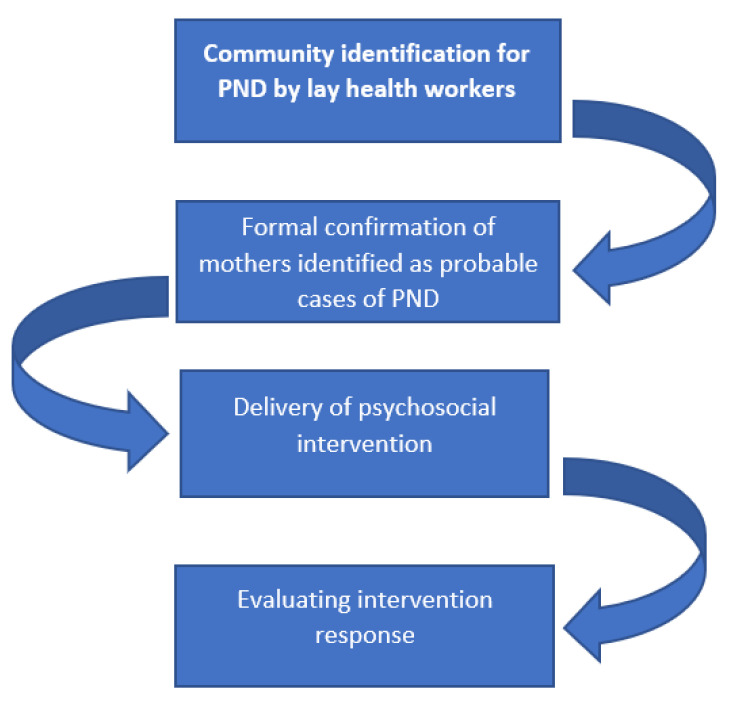
Approach for phased identification and screening for perinatal depression in Pakistan.

**Figure 2 ijerph-18-06693-f002:**
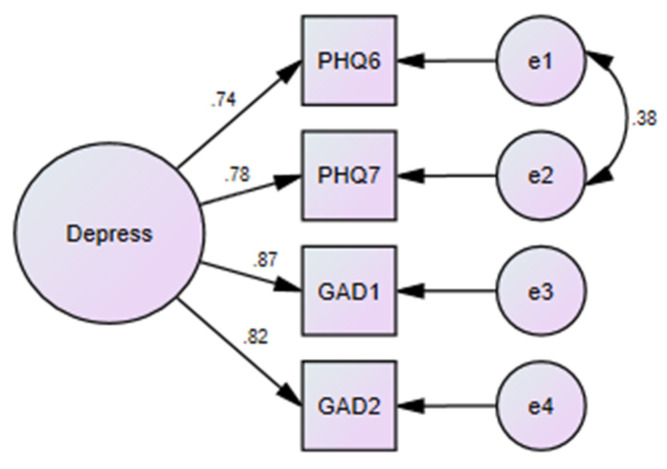
Model fit for the PHQ-4 as per confirmatory factor analysis.

**Figure 3 ijerph-18-06693-f003:**
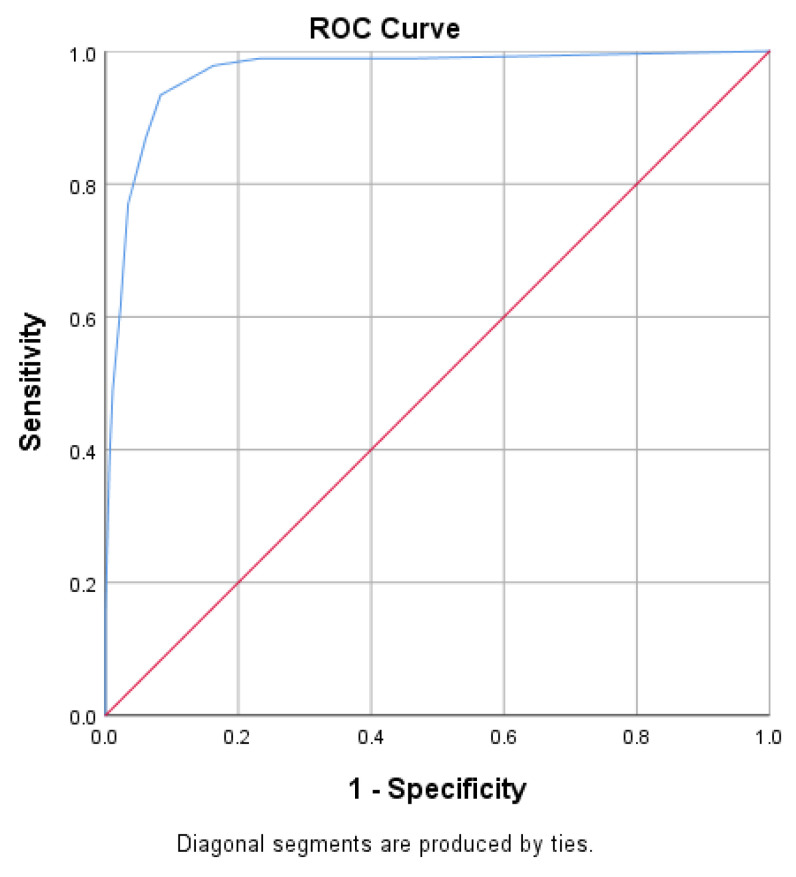
ROC curve for the PHQ-4.

**Figure 4 ijerph-18-06693-f004:**
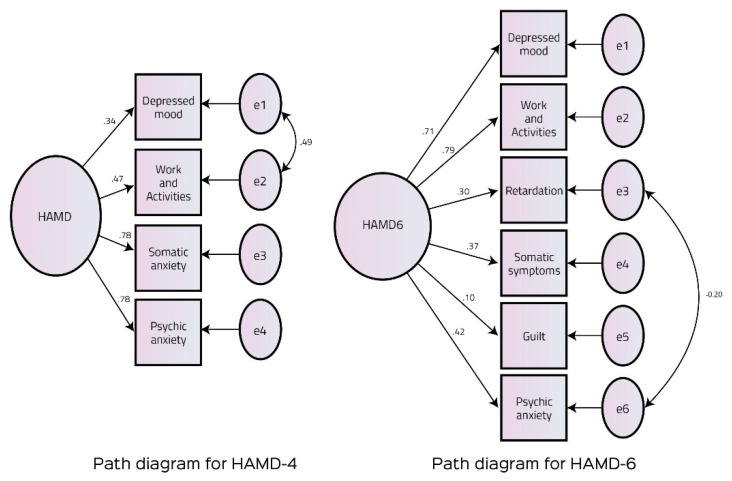
Path diagrams for the HAMD-4 and HAMD-6.

**Figure 5 ijerph-18-06693-f005:**
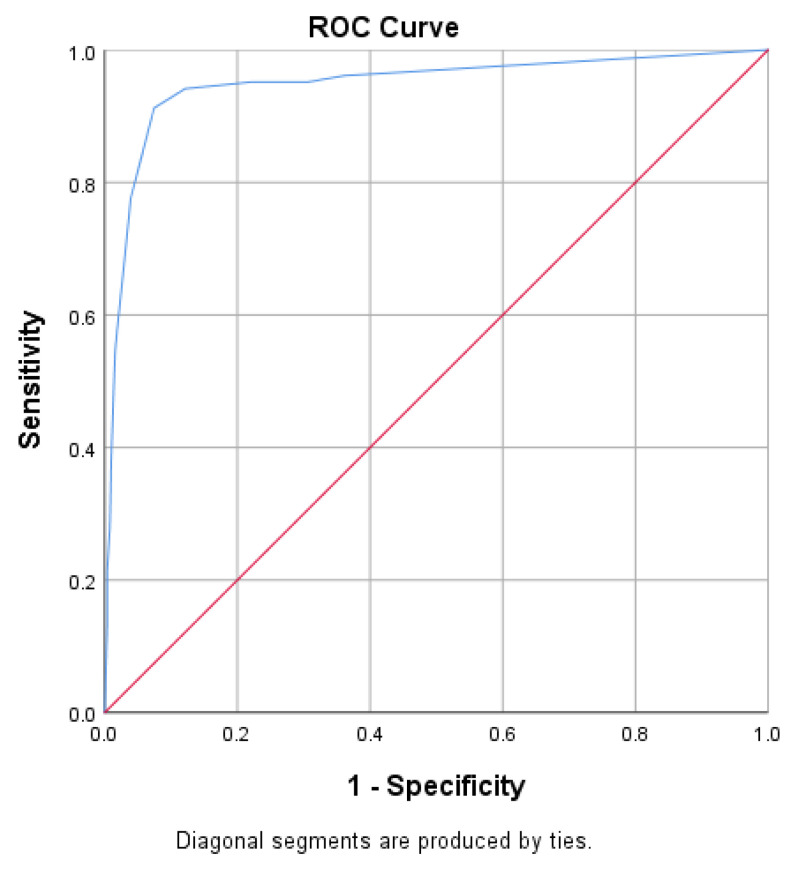
ROC curve for the HAMD-4.

**Table 1 ijerph-18-06693-t001:** Proposed tools for screening and response monitoring for perinatal depression.

Step	Scale	Items in the Scale	Delivery Agent
Phase 1: Identification of women at high risk of perinatal depression	Community informant detection tool	4 social stressors 9 symptoms of perinatal depression Visual analog scale	(1) Community health workers (2) Peers
Phase 2: Screening of high-risk women for perinatal depression	Patient Health Questionnaire-4 items	Two depressive symptoms assessed using a Patient Health Questionnaire (2 items) Two anxiety symptoms assessed using a Generalized Anxiety Disorder Scale (2 items)	Lady Health Workers employed at Primary Care Centers
Phase 3: Measuring treatment responses	Hamilton Depression Rating Scale-4 items	Four symptoms derived from four items on the Hamilton Depression Rating Scale Depressed mood, anhedonia, psychic anxiety, and somatic anxiety	Lady Health Workers employed at Primary Care Centers

**Table 2 ijerph-18-06693-t002:** Fit indices for the HAMD-6 and HAMD-4.

Metric for Goodness of Fit	HAMD-6	HAMD-4
Chi-square (*p*-value)	43.06 (<0.001)	4.30 (0.04)
CMIN/df	5.38	4.30
GFI	0.985	99.8
AGFI	0.960	97.6
CFI	94.9	99.6
TLI	90.4	97.9
NFI	93.8	99.5
RMSEA	0.07	0.06
PCLOSE	0.05	0.28
AIC	69.06	22.30
BIC	131.54	65.55

## Data Availability

The datasets are available from the corresponding author on request.

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
