# Peer review of "Scalable Screening and Treatment Response Monitoring for Perinatal Depression in Low- and Middle-Income Countries"

_ijerph, 2021, doi:10.3390/ijerph18136693_

Round 1

Reviewer 1 Report

The question of how to scale up identification of common perinatal mental disorders in LMIC is a very important one, and the present manuscript aims to address this question. The feasibility and psychometric properties of each of the three short tools (CIDT, PHQ4 and HAMD4) are appropriately presented but some have already been published elsewhere and it is not clear how this all ties together. It may help to set the reader's expectations by stating at the outset that this paper is a secondary analysis summary of psychometric properties of the three short tools, and a proposal for how to use them that needs to be further tested.

The authors do make a case for the need for a multi step screening process and the one they have laid out seems appropriate in theory but there needs to be more of a justification for each of the steps proposed. For example, the CIDT with 9 items, modified for cultural appropriateness and linguistic abilities, does seem to mirror the PHQ9 in terms of number of items, and content assessed. So it would seem that a head to head comparison of CIDT and PHQ9 is in order - perhaps a CIDT does not need to be followed with a PHQ9 or a PHQ4 thereby eliminating one step?

Although the authors state correctly that there is a need to test the implementation of screening tools in real world settings, the setting within which they have tested their approach is more of an ideal setting, with ongoing implementation of mhGAP, THP, and infrastructure for multiple RCTs.

The literature is inaccurately cited in places. For example, Zlotnick et al did not "find multi step approach of screening for perinatal depression practical" Rather this was an RCT of prevention of PPD, the risk index questionnaire was for inclusion of participants in the RCT and the BDI was an outcome measure. Similarly, Yawn et al actually found that "the two step assessment was not feasible for the usual care group".

This is also demonstrated in a plethora.....this paragraph needs to be edited for clarity.

Author Response

Please, find detailed responses to all the comments in the word file attached. 

Reviewer 2 Report

The paper under review addresses a gap in clinical care: namely by testing shorter tools that would be appropriate for screening and for monitoring treatment response. Findings provide psychometric support for the use of three measures: CIDT, PHQ4, HAMD-4, each in a specific capacity: risk assessment, high risk assessment, and treatment monitoring, respectively.

Generally, I found the manuscript to be well written, the flow of information was easy to follow and provided enough detail so that findings were easy to understand. I also found the rationale for the study to be compelling. Therefore, my feedback for revisions is relatively minimal and mainly focused on clarifying details, rather than for major restructuring or rewriting of the paper.

Introduction:

  1. Authors introduce at the end the introduction section the EPDS scale – what does the acronym stand for? Because authors introduce it here, I was left wondering why this wasn’t one of the scales tested? Are the authors stating that this is in development and being tested already and authors wanted to test existing scales? Perhaps add another sentence or two here to clarify goals.
  2. I would list study aims as a separate paragraph.
  3. Aim a) refers to two phases, but later in the paper a third phase is being referenced. Please clarify.
  4. Perhaps authors can split phases up and more clearly describe up front what each phase entails.

Methods:

  1. In Approach authors again describe two phases but then three phases are described later in the methods section. Please clarify/rectify. I would keep the three phase organization.
  2. Under Settings:
    1. what does WHO mhGAP stand for? Please spell out.
    2. In this section authors also describe three datasets. Are these datasets each associated with a study phase?
    3. I am wondering if this paragraph would be better suited for the Current Study section at the end of the introduction. I will leave it to the authors to decide.
  3. Phase 1, last paragraph:
    1. What does CHW stand for?
    2. The sample sizes referred to in this paragraph are unclear as to what they represent – are these participants or recruiters? The results listed in the first paragraph of the Results section refer to 435 women for this part of the study so it is unclear where the discrepancy is coming from. Please rectify/clarify.
  4. I found the description and division of information in the three stages helpful from a reader perspective.  
  5. Establishing validity of these scales comes up at the end of the methods section in Statistical Analyses. I would mention it as a goal of the study from the beginning in the introduction since it is a strong feature of this study.

Results:

  1. I would encourage the authors to use the same Phase delineation and naming throughout the results section too. The title of the first subsection is different from the title of Phase 1 in the Methods so it is a bit confusing. Please stay consistent with naming to help readers move through this paper easily.
  2. I noticed some variability in the demographic data provided for each sample. Was it collected like that or could there be consistency in what data are provided? For example, in section one authors present salaries, while in section two and three they present whether women were part of joint households.
  3. Before providing the psychometric properties, I would appreciate some basic descriptive information about each of the measures to understand the clinical makeup of each of the samples, such as mean (SD), median, and range of scores on each of the scales. This could easily be done in a single statement for each scale in part.

Discussion:

  1. The second paragraph in the discussion section seems better fit in the introduction since it is all about prior work.
  2. Use the space here to discuss findings and compare to prior findings. For example, I would discuss how sensitivity and specificity in current findings compare to these prior studies on longer scales.
  3. The discussion appears to have been dedicated to reiterating the importance of the study. I would use this space instead to take the findings from each of the three phases in turn and discuss each relative to the literature and findings for similar scales. Then findings can be put into the larger context by detailing the implications for mental health. The way the discussion is currently written, I am not sure as a reader what to make of all the important reliability and validity findings. In other words, what does it mean that reliability and validity were excellent on the whole? The goal is to help the readers make sense of this work within the context of the literature.

Author Response

Please, find attached detailed responses to reviewers' comments in the word file attached herewith.
